# UniVIEDM: A Diffusion Model to Unify Visual Information Extraction Subtasks

## Abstract

Visual Information Extraction (VIE) focuses on extracting named entities and their relationships from visually rich document images. Traditionally, VIE systems rely on three separate models to handle three distinct subtasks, but the emerging trend in research is to design a single model that can address all of these tasks simultaneously. However, current methods face quadratic computational complexity when extracting entity relationships, as they must iterate over all token pairs. To address this issue, this paper introduces a Unified VIE Diffusion Model (UniVIEDM) for all tasks within VIE. UniVIEDM generates entity labels and their relationships conditioned on their plane coordinates, greatly reducing the computational complexity. UniVIEDM represents the layout of each visually rich document as a plane graph and converts the three subtasks into plane graph generation problems. During the pre-training stage, UniVIEDM leverages a jump-diffusion process to learn to generate valid sets of bounding boxes for all words and line segments connecting different boxes. During the fine-tuning stage, UniVIEDM employs a continuous-time Markov chain diffusion model to learn to predict the labels of boxes and line segments based on their coordinate features.

## 1 Introduction

The visual information extraction task (VIE) involves extracting texts of multiple key segments from given document images and saving these texts to structured documents. With the acceleration of the digitization process, the VIE task has been regarded as a crucial part of intelligent document processing and is required by many real-world applications in various industries such as finance, medical treatment, and insurance Cui (2021). A VIE task is often divided into three sub-tasks, namely line grouping, semantic entity recognition (SER) and relation extraction (RE) Li et al. (2021b); Zhang et al. (2021); Lin et al. (2024), the illustration of three sub-tasks can be found in Figure 1. Line grouping aims to combine different words into text fields, SER aims to determine the semantic labels of different text fields, and RE aims to identify the key-value pair relationships between text fields. The past decades have witnessed the progress of VIE Gu et al. (2022); Xu et al. (2020; 2021a); Huang et al. (2022); Xu et al. (2021b); Li et al. (2021a); Appalaraju et al. (2021). However, since these three subtasks are modeled separately, the VIE systems are overly complex. Therefore, one emerging trend in research is to design a single model that can address all of these tasks simultaneously. In this paper, we aim to unify the three subtasks at the levels of task formulation, modeling approach, and inference model. This unification allows the system to perform all tasks using a single model, simplifying the overall structure and reducing computational complexity greatly.

Current approaches for developing general models for VIE rely on pre-trained multimodal encoders to learn robust features for tokens and their pairs, and then determine labels based on these features. Notable systems include ESP Yang et al. (2023), PEneo Lin et al. (2024), and UniVIE Hu et al. (2024). For instance, the UniVIE model constructs an $N \times N$ token pair matrix, using the diagonal elements to predict token labels and the off-diagonal elements to predict inter- or intra-entity relationships. These approaches typically encounter two significant challenges. First, they face scaling issues, as the size of the token pair matrix increases quadratically with the token length $N$. Second, while they unify the task formulation across the three subtasks, they fail to unify the underlying mathematical principles, resulting in substantial design differences between the neural networks used for each subtask. In summary, the question of how to unify the three subtasks at the levels of task formulation, modeling approach, and inference model remains an open challenge.

Figure 1: Illustration of the objectives and forms of the three subtasks. (a) Line grouping task, which combines different words into text fields. (b) SER task, which determines the semantic labels of different text fields. (c) RE task, which identifies the key-value pair relationships between text fields. The sequential execution of these three subtasks leads to the error propagation probelm.

To overcome these limitations, this paper introduces the concept of plane graphs to model layout modality information, transforming the three subtasks (line clustering, SER, and RE) into plane graph generation problems. As illustrated in Figure 2, each word's bounding box is represented as a "node" in the graph, and the relationships (such as being on the same line or forming a key-value pair) between pairs of words are represented as "edges" in the graph. Plane graphs embed "nodes" and "edges" in a plane, allowing them to be described using natural coordinates. This enables connectivity between "nodes" to be determined based on the natural coordinates of "nodes" and "edges". Plane graphs differ from general graphs by using natural coordinates of "nodes" and "edges" (continuous random variables) to describe connectivity, rather than relying on an "adjacency matrix" (discrete random variables). This approach provides a unified data format, allowing line grouping and RE to be solved based on the natural coordinates of "edges", just as SER is based on the natural coordinates of "nodes". The introduction of the plane graph concept unifies the modeling methods of the three subtasks, allowing them to be completed as an integrated whole.

Meanwhile, it is non-trivial to learn a valid generative model for plane graphs. First, because the number of nodes (edges) is not fixed, the collection of coordinate data for all nodes (edges) forms "trans-dimensional data," meaning the data dimension varies across samples. Because the subtraction between two trans-dimensional data is not well defined, the score function in diffusion models that excel at modeling "fixed-dimensional data," such as DDPM Ho et al. (2020), is not applicable. Second, because the probability of a plane graph does not depend on the order of nodes (or edges), auto-regressive models like transformers are not well-suited for this task. Third, since the variables in a plane graph include both continuous (coordinates) and discrete (labels) elements, the generative model must be capable of handling mixed-type random variables.

This paper proposes **UniVIEDM**, a unified layout model capable of addressing all tasks within VIE. UniVIEDM incorporates two continuous-time diffusion processes: the first is based on a jump-diffusion process, and the second on a Continuous Time Markov Chain (CTMC). In the jump-diffusion process, UniVIEDM learns to generate valid coordinates for nodes (edges) and produces coordinate features. In the CTMC process, UniVIEDM generates the labels for both nodes and edges, conditioned on the coordinate features and text features, simultaneously completing the tasks of line grouping, SER, and RE. During the jump-diffusion process, a new node (or edge) is added to the existing plane graph with a small probability, which is controlled by a node (or edge) number prediction network. The coordinates of the new node (or edge) are predicted by a transformer network. Throughout this process, a coordinate-denoising graph neural network continuously modifies the coordinates of existing nodes (or edges). During the CTMC process, a label-denoising graph neural network predicts the label of each node (or edge) based on the coordinate and text features.

Our contributions are summarized as follows:

- UniVIEDM unifies the three subtasks of VIE. The benefit of UniVIEDM lies in easing the challenge of quadratic computational complexity.

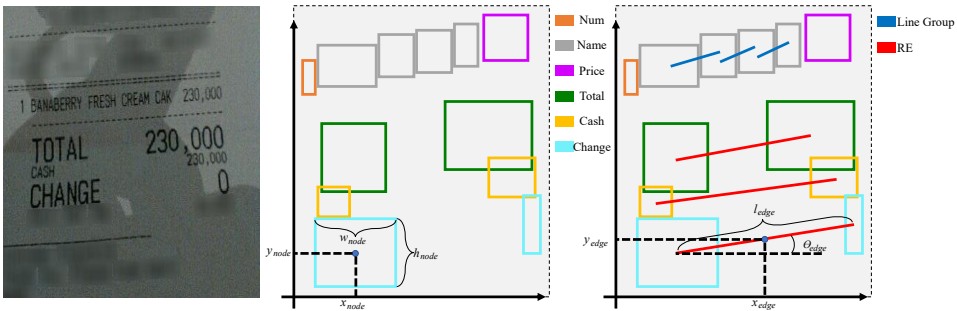

Figure 2: Plane graph of a sample. Receipt (left), nodes of the graph (center), edges of the graph (right). Better viewed in color.

- With the help of jump-diffusion and CTMC, UniVIEDM successfully addresses the challenges of varying data dimensions and insensitivity to the generation order.
- UniVIEDM achieves SOTA or competitive performance on several public datasets.

## 2 RELATED WORKS

### 2.1 UNIFIED VIE MODELS

The challenge of unifying all three tasks within VIE has attracted significant attention from researchers Zhang et al. (2020); Wang et al. (2021); Cheng et al. (2022); He et al. (2023); Yang et al. (2023); Lin et al. (2024); Hu et al. (2024). Both generative and discriminative models are viable technical approaches. While the discriminative approach has been extensively researched, the generative approach has not. In the early days, discriminative models like TRIE Zhang et al. (2020), VIES Wang et al. (2021), and TRIE++ Cheng et al. (2022) proposed to integrate the OCR stage with SER. However, their drawback lies in the poor generalization ability to different downstream tasks. Later, ESP Yang et al. (2023) replaced the OCR stage with a ConvNeXt-FPN backbone to detect text and output visual feature maps. The SER and RE modules are sequentially concatenated after the ConvNeXt-FPN backbone. However, ESP overlooks the error accumulation problem because the SER and RE modules are processed sequentially. PEneo Lin et al. (2024) and UniVIE Hu et al. (2024) both constructed an $N$ by $N$ token pair matrix to perform the three subtasks. Inspired by the TPLinker Wang et al. (2020) method, PEneo constructed five token pair matrixes and fed them into three different layers to accomplish line extraction, line grouping, and entity linking. UniVIE used the diagonal matrix elements to predict the labels of tokens and the off-diagonal elements to predict the inter- or intra-entity relationships. Both PEneo and UniVIE face scaling issues because the size of the token pair matrix grows quadratically with the token length $N$.

State-of-the-art generative models encompass autoregressive models, such as LLMs Yenduri et al. (2024), and diffusion models, like DDPM Ho et al. (2020). ICL-D3IE He et al. (2023) first applies OCR to extract the contents and coordinates of words. Then, it fine-tunes a large language model (LLM) to predict entity labels and spatial relationships between entities. Rather than explicitly modeling the layout modality, ICL-D3IE requires the LLM to determine spatial relationships by directly comparing the coordinate values. In contrast, this paper proposes an explicit diffusion layout model that addresses the three subtasks from a unified perspective.

### 2.2 LAYOUT MODELS

A layout model is defined as a non-sequential collection of mixed-type random variables (coordinates and labels) and their joint distribution. Modeling the layout modality is a challenging problem that has captured the attention of many researchers Li et al. (2019); Jyothi et al. (2019); Zheng et al. (2019); Lee et al. (2020); Patil et al. (2020); Gupta et al. (2021); Arroyo et al. (2021); Yamaguchi (2021); Kikuchi et al. (2021); Jiang et al. (2022); Kong et al. (2022); Chai et al. (2023). Their problem setup has been generating sets of box coordinates given sets of labels. Researchers have focused on addressing the challenges of varying data dimensions and insensitivity to the generation order.

Figure 3: Framework of UniVIEDM. Colored dots and line segments form the plane graph in Figure 2. UniVIEDM starts with the diffusion of node coordinates followed by edge coordinates. Then the nodes and edges are processed to obtain d-dimensional features. At last, The random label sets are used in both the Node Labels Diffusion Module and the Edge Labels Diffusion Module, with each stage generating predicted label sets. Better viewed in color.

Autoregressive models are effective at generating an arbitrary number of boxes. This approach converts the sets of coordinates into a sequence and then designs backbones that excel at modeling sequential data. Typical models like Layout-Transformer Gupta et al. (2021), BLT Kong et al. (2022), and Layout-DM Chai et al. (2023) first use discrete tokens to replace the continuous coordinates and then concatenate the coordinate and label tokens to form the target sequence. However, autoregressive models struggle with the generation order problem due to their sequential nature.

Autoencoder models are typically effective at addressing the problem of generation order. This approach usually first samples a certain number of random box coordinates and then iteratively modifies them. Models such as Layout-GAN Li et al. (2019), NDN Lee et al. (2020), and VTN Arroyo et al. (2021) employ different neural network architectures, including transformers and GNNs. However, autoencoders do not consider the impact of the number of boxes during generation.

A promising direction is to combine the strengths of variational autoencoders and autoregressive models. In the Layout-VAE Jyothi et al. (2019), a Count-VAE predicts the number of boxes for each label. Based on the label sets and the number of boxes, a Bbox-VAE generates the continuous coordinates of each box one-by-one. However, Layout-VAE suffers from a weak encoder-decoder problem because denoising diffusion models have been empirically shown to outperform VAE encoder-decoder Chen et al. (2024).

## 3 UNIFIED VIE DIFFUSION MODEL

### 3.1 PROBLEM SETUP

As shown in Figure 2, a plane graph consists of $N$ nodes and $M$ edges. Each node represents a bounding box of a word output by the OCR engine, and each edge represents a line segment linking two bounding boxes. Each node (or edge) has a continuous 4-dimensional coordinate vector and a discrete label. The coordinates of the $i$th node are denoted as $\mathbf{x}_i^{node} := \left(x_{node}^i, y_{node}^i, w_{node}^i, h_{node}^i\right)$, where $x_{node}^i$ and $y_{node}^i$ indicate the position of the box, and $w_{node}^i$ and $h_{node}^i$ describe the width and height of the box. The coordinates of the $j$th edge are denoted as $\mathbf{x}_j^{edge} := \left(x_{edge}^j, y_{edge}^j, l_{edge}^j, \theta_{edge}^j\right)$ where $l_{edge}^j$ is the length and $\theta_{edge}^j$ is the angle. Labels of nodes and edges are denoted as $\mathbf{z}^{node}$ and $\mathbf{z}^{edge}$. There are $S$ types of node labels and 2+ types of edge labels. When $\mathbf{z}^{edge} = 0$, it indicates that two boxes should be merged into one in the line grouping task; otherwise, they have a relationship in the RE task.

This paper uses the upper case letter $\mathbf{X}$ to denote one plane graph sample, which is defined as the collection of all $N$ nodes and $M$ edges, i.e.,

$$\mathbf{X} := \left\{\mathbf{x}_i^{node}, \mathbf{z}_i^{node}\right\}_{i=1,\cdots,N} \bigcup \left\{\mathbf{x}_j^{edge}, \mathbf{z}_j^{edge}\right\}_{j=1,\cdots,M}.$$

A plane graph $\mathbf{X}$ lives in a trans-dimensional sample space, i.e., for two different samples $\mathbf{X}_A$ and $\mathbf{X}_B$, it is very likely that $N_A \neq N_B$ and $M_A \neq M_B$. According to the theories of spatial point process Baddeley et al. (2013), a well-defined probability distribution can be defined over this trans-dimensional sample space such that the probability of sampling a particular plane graph $\mathbf{X}$ makes

sense. This paper uses $P(\mathbf{X})$ to represent this probability and assumes it can be factorized as

$$P(\mathbf{X}) = \tag{1}$$

$$P\left(\left\{\mathbf{x}_i^{node}, \mathbf{z}_i^{node}\right\}_{i \leq N} \bigcup \left\{\mathbf{x}_j^{edge}, \mathbf{z}_j^{edge}\right\}_{j \leq M}\right) = \tag{2}$$

$$P\left(\left\{\mathbf{x}_i^{node}\right\}_{i \leq N}\right) * \tag{3}$$

$$P\left(\left\{\mathbf{x}_j^{edge}\right\}_{j \leq M} \middle| \left\{\mathbf{x}_i^{node}\right\}_{i \leq N}\right) * \tag{4}$$

$$P\left(\left\{\mathbf{z}_i^{node}\right\}_{i \leq N} \middle| \left\{\mathbf{x}_i^{node}\right\}_{i \leq N}, \left\{\mathbf{x}_j^{edge}\right\}_{j \leq M}\right) * \tag{5}$$

$$P\left(\left\{\mathbf{z}_j^{edge}\right\}_{j \leq M} \middle| \left\{\mathbf{x}_i^{node}\right\}_{i \leq N}, \left\{\mathbf{x}_j^{edge}\right\}_{j \leq M}\right). \tag{6}$$

This section aims to learn the probability (2) by fitting the four quantities in equations (3)-(6). Since the equation (3) does not require any labels, it corresponds to a pre-training task of reconstructing all node coordinates of training data. The equation (5) corresponds to predicting the node labels ( the SER task). The equation (4) generates all edge coordinates. The equation (6) generates labels of all edges (the line grouping and RE task).

## 3.2 The Framework of UniVIEDM

Inspired by equations (3)-(6), the framework of UniVIEDM consists of 4 modules. As shown in Figure 3, 4 modules are organized into three stages. In the first stage, UniVIEDM trains a node coordinate diffusion module to fit the probability distribution of node coordinates on the plane. It then encodes the observed node coordinates to obtain their feature representations. The edge coordinate diffusion module generates the edge coordinates, conditioned on node coordinate features.

The second and third stages correspond to the node and edge label generation modules, which operate simultaneously. Each module uses the feature representations from the node and edge coordinate diffusion modules as conditions for label generation. Starting from random label sets, they employ diffusion models designed for discrete variables to predict the labels, conditioned on the feature representations of the corresponding spatial coordinates.

## 3.3 Coordinates Denoising Based on Jump-Diffusion

This paper adopts the jump-diffusion process for the forward and backward processes. The mathematics of a jump-diffusion process can be found in Campbell et al. (2024). Over time, a jump-diffusion process can involve two types of events: jump events and diffusion events. Jump events occur at specific moments, changing the number of nodes (or edges) by either adding or removing one (a node/edge is randomly deleted in the forward process, while a new node/edge is added in the reverse process). Diffusion events happen during the remaining time, continually adjusting the coordinates of each node (or edge) based on the current coordinates (Gaussian noise is added to each node/edge's coordinates in the forward process, and a portion of this noise is removed in the reverse process). This approach enables the continuous coordinate diffusion algorithm to flexibly determine and generate the appropriate number of edges and their coordinates, based on the observed node coordinates.

Let $\mathbf{X}_t$ represents only $\left\{\mathbf{x}_i^{node}\right\}_{i=1,\cdots,N}$ temporarily, and $\mathbf{Y}_{del}$ represents $\left\{\mathbf{x}_i^{node}\right\}_{i=1,\cdots,N-1}$, then the forward process is controlled by the following equations

$$\textbf{Jump}: \mathbf{X}_t' = \begin{cases} \mathbf{X}_t, & with\ prob\ 1 - \lambda(n_t) \\ \mathbf{Y}_{del}, & with\ prob\ \lambda(n_t) \end{cases} \tag{7}$$

$$\textbf{Diff}: \mathbf{x}_{t+dt} = \mathbf{x}_t' - \frac{\beta_t}{2}\mathbf{x}_t' dt + \sqrt{\beta_t} d\mathbf{w}_t \tag{8}$$

where $\lambda(n_t)$ represents the jumping rates and $\lambda(n_t = 1) = 0$. $\beta_t$ represents the noise schedule reported in DDPM.

To learn the reverse process, three neural networks are needed. They are the coordinates denoising GNN $s_t^\theta(\mathbf{X}_t)$, new coordinate autoregressive network $A_t^\theta(\mathbf{y}_i^{add}|\mathbf{X}_t)$, and node number prediction network $\overline{\lambda}_t^\theta(\mathbf{X}_t)$. Their network design can be found in the appendix. The corresponding reverse process is then controlled by the following equations.

$$\mathbf{Jump} : \mathbf{X}_t' = \begin{cases} \mathbf{X}_t, & with\ prob\ 1 - \overline{\lambda}_t^\theta(\mathbf{X}_t) \\ \mathbf{Y}_{add}, & with\ prob\ \overline{\lambda}_t^\theta(\mathbf{X}_t) \end{cases} \tag{9}$$

$$and\ \mathbf{Y}_{add} = \left[\mathbf{X}_t; \mathbf{y}_i^{add}\right],\ \mathbf{y}_i^{add} \sim A_t(\mathbf{y}_i^{add}|\mathbf{X}_t) \tag{10}$$

$$\mathbf{Diff} : \mathbf{x}_{t+dt} = \mathbf{x}_t' - \left[\frac{\beta_t}{2}\mathbf{x}_t' + s_t^\theta(\mathbf{X}_t)\right] dt + \sqrt{\beta_t}d\mathbf{w}_t \tag{11}$$

Based on the features output by the node networks, the coordinates of edges are modeled using the same equations listed in (7)-(11).

## 3.4 LABELS DENOISING BASED ON CTMC

Because the coordinate diffusion modules are based on continuous-time processes, methods like D3PM Austin et al. (2021), which rely on discrete-time processes, are not compatible. CTMC stands out as an excellent alternative. In contrast to discrete-time processes, where state transitions are limited to specific time intervals, continuous-time processes enable state changes to happen at any given moment. In these processes, the system continuously moves from one state to the next, following a random period spent in the current state. A CTMC process is fully controlled by the initial distribution of data and its transition rate matrix

$$R_t(\mathbf{z}_{t+h}, \mathbf{z}_t) := \lim_{h \to 0} \frac{P\left(\mathbf{z}_{t+h}|\mathbf{z}_t\right) - \delta_{z_{t+h}, z_t}}{h} \tag{12}$$

Therefore, the infinitesimal transition probabilities of a CTMC can be defined by the following expansion

$$P\left(\mathbf{z}_{t+h}|\mathbf{z}_t\right) = \delta_{z_{t+h}, z_t} + R_t(\mathbf{z}_{t+h}, \mathbf{z}_t)h + o(h) \tag{13}$$

A proper transition rate matrix $R_t$ is designed such that the time reversal of this CTMC is also a CTMC with the following backward transition rate matrix $\hat{R}_t$ The forward and backward rate matrix are related by the following equations

$$\hat{R}_t(\mathbf{z}_1, \mathbf{z}_2) = R_t(\mathbf{z}_2, \mathbf{z}_1)\frac{P_t(\mathbf{z}_2)}{P_t(\mathbf{z}_1)} \tag{14}$$

Since the probability ratio $\frac{P_t(\mathbf{z}_2)}{P_t(\mathbf{z}_1)}$ in the above equation is unknown, this paper designs a neural network to approximate it. This network boils down to predicting the initial state given $\mathbf{z}_{t=0}$ given the noised labels $\mathbf{z}_t$.

## 3.5 TRAINING AND INFERENCE

According to the research in Campbell et al. (2024), our training loss of coordinates diffusion includes 3 terms. The first term minimizes the score-matching loss of coordinates of existing nodes. The second term minimizes the jumping-rate loss. The third term minimizes the cross-entropy loss of predicted initial labels. The label diffusion loss is the continuous time elbo derived in Campbell et al. (2022).

During testing, UniVIEDM first uses the node coordinates denoising network to extract features. Then the edge coordinates denoising network will generate the coordinates of edges in the line grouping task and RE task. At last, the predictor-corrector Campbell et al. (2022) sampling method is used to determine the labels of both nodes and edges based on their coordinates features.

Figure 4: Qualitative results of node label diffusion on the SIBR Dataset. (a) Generated samples. (b) A sample path of the diffusion process. In (a), UniVIEDM's predictions for four label categories in the SIBR dataset are shown, with different colors representing each label. All entity labels are predicted accurately. In (b), the final few jumps of the reverse jump diffusion process are depicted, with the rightmost image showing the final prediction. Before reaching this stage, each jump modifies the label of only one box. The boxes filled with color highlight where the label change occurred.

Table 1: Realistic VIE performance on SIBR. The baseline results are directly cited from the findings reported in the literature. "SER+RE" means all sub-tasks are solved sequentially."Joint" means all sub-tasks are solved jointly. "Image2Seq" means the models are OCR-free. "RE F1" means ground truth labels of SER are used, while "Pair F1" means predicted labels of SER are used. "Layout embedding only" means the feature of coordinates output by UniVIEDM.

| Model | Reference | Description | | Performance | | |
|---|---|---|---|---|---|---|
| | | Venue | Pipeline | SER F1↑ | RE F1↑ | Pair F1↑ |
| ESP | Yang et al. (2023) | CVPR'23 | Image2Seq | 95.27 | 85.96 | - |
| UniVIE | Hu et al. (2024) | ICDAR'24 | Joint | 96.68 | 87.72 | - |
| Donut Base | Kim et al. (2022) | ECCV'22 | Image2Seq | - | - | 17.26 |
| LayoutXLM | Xu et al. (2021b) | arXiv'21 | SER+RE | 93.61 | 81.99 | 70.45 |
| LiLT-InfoXLM Base | Wang et al. (2022) | ACL'22 | SER+RE | 92.90 | 89.00 | 72.76 |
| LayoutLMv3 Chinese Base | Huang et al. (2022) | ACMMM'22 | SER+RE | 93.50 | 87.07 | 73.51 |
| PEneo-LayoutLMv3 Chinese Base | Lin et al. (2024) | ACMMM'24 | Joint | - | - | 82.52 |
| Ours (Layout embedding only) | - | ICLR'25 | Joint | 83.58 | - | 68.75 |
| Ours (Word2vec embedding) | - | ICLR'25 | Joint | 91.59 | - | 80.80 |
| Ours (Byte pair encoding embedding) | - | ICLR'25 | Joint | 93.22 | - | 81.92 |
| Ours (LayoutLMv3 Chinese Base) | - | ICLR'25 | Joint | 96.86 | - | 83.42 |

## 4 EXPERIMENTS

In this section, we evaluate the performance of UniVIEDM's four modules using public datasets. Specifically, we assess the realistic VIE performance on the SIBR dataset. We also evaluate the effectiveness of the coordinate diffusion modules in generating plausible results on the CORD dataset, along with the accuracy of the label diffusion modules under different feature conditions on the SIBR dataset. Further details on UniVIEDM's training and implementation are provided in the appendix.

### 4.1 PUBLIC DATASETS

We conduct experiments on CORD Park et al. (2019),SIBRYang et al. (2023). The CORD dataset is a widely used public benchmark, comprising 800 training images, 100 validation images, and 100 test images. The dataset spans 30 different fields, such as item names, quantities, and total amounts, and presents information in a complex, hierarchical structure with nested groups. The SIBR dataset is bilingual, containing 600 Chinese invoices, 300 English bills of entry, and 100 bilingual receipts, amounting to 600 training samples and 400 test samples. It provides line-level annotations, including entity linking between lines and line grouping within individual forms.

### 4.2 IMPLEMENTATION DETAILS

We employ AdamW as the optimizer. The learning rate is set to 1e-6. We pre-trained the layout reconstruction capability of the spatial denoising network using unlabeled OCR text boxes across all training datasets. A total of 10,000 batches were trained, with each batch containing 130 samples. Subsequently, we froze the spatial denoising network and trained the edge coordinate denoising

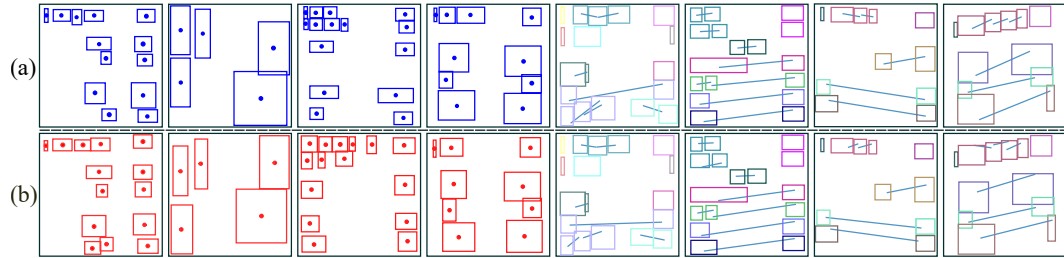

Figure 5: Qualitative results of node and edge coordinate diffusion on the CORD dataset. (a) Ground truth samples. (b) Generated samples. The left four columns display the diffusion results for node coordinates. Since the corresponding samples in (b) are unconditionally generated, they do not exactly match their counterparts in (a), but exhibit similar layouts. The right four columns show the diffusion results for edge coordinates. As the corresponding samples in (b) are conditionally generated, they can be directly compared to the annotated results in (a).

Table 2: Realistic VIE performance test on CORD. The baseline results are directly cited from the findings reported in the literature. "SER+RE" means all sub-tasks are solved sequentially. "Joint" means all sub-tasks are solved jointly. "Image2Seq" means the models are OCR-free. "RE F1" means ground truth labels of SER are used, while "Pair F1" means predicted labels of SER are used. "Layout embedding only" means the feature of coordinates output by UniVIEDM.

| Model | Reference | Description | | Performance | | |
| | | Venue | Pipeline | SER F1↑ | RE F1↑ | Pair F1↑ |
|---|---|---|---|---|---|---|
| ESP | Yang et al. (2023) | CVPR'23 | Image2Seq | 95.65 | 98.8 | - |
| BERT Base | Kenton & Toutanova (2019) | NAACL'19 | SER+RE | 89.68 | 92.83 | - |
| BROS Base | Hong et al. (2022) | AAAI'22 | SER+RE | 96.5 | 95.73 | - |
| RoBERTa Base | Cui et al. (2020) | arXiv'20 | SER+RE | 93.54 | - | - |
| LayoutLM Base | Xu et al. (2020) | SIGKDD'20 | SER+RE | 96.07 | 95.21 | - |
| LayoutLMv2 Base | Xu et al. (2021a) | arXiv'21 | SER+RE | 94.95 | 95.59 | - |
| LayoutLMv3 Large | Huang et al. (2022) | ACMMM'22 | SER+RE | 97.46 | 98.28 | - |
| DocFormer Large | Appalaraju et al. (2021) | ICCV'21 | SER+RE | 96.99 | - | - |
| Ours (Layout embedding only) | - | ICLR'25 | Joint | 88.73 | - | 90.31 |
| Ours (Word2vec embedding) | - | ICLR'25 | Joint | 91.36 | - | 96.67 |
| Ours (Byte pair encoding embedding) | - | ICLR'25 | Joint | 93.44 | - | 96.67 |
| Ours (LayoutLMv3 Chinese Base) | - | ICLR'25 | Joint | 97.11 | - | 96.67 |

network and label denoising network using labeled data from three sub-tasks. For text modality perception, we experimented with both static and dynamic word embeddings. We employed Word2Vec for static embeddings and the LayoutLMv3 model series for dynamic embeddings.

## 4.3 REALISTIC VIE PERFORMANCE TEST ON SIBR

**Evaluation metrics.** In prior works such as LayoutLMv3, RE tasks are evaluated by inputting the ground truth from line clustering and SER directly into the RE module. This approach, however, neglects the issue of error propagation from earlier tasks, resulting in artificially higher performance metrics compared to the system's true capabilities. To address this, studies like PEneo Lin et al. (2024) take error accumulation into account, offering more realistic performance evaluations. In this section, we adopt the testing methodology used in PEneo. To rigorously evaluate UniVIEDM's RE performance, we designed a post-processing step for the edge coordinate diffusion process. For each generated edge, we first calculate the coordinates of its two endpoints and then identify the nearest node to each. The edge is considered to predict a relationship between these two nodes. We discard edges if they connect the same node or if the endpoints are too far from any nodes. For the SIBR dataset, the text labeled as "answer" is always concatenated after the text labeled as "question." The concatenated texts are then used to compute the pair-level F1-score.

**Quantitative and qualitative results.** As shown in Table 1, our model achieves the highest SER F1 score of 96.86%, surpassing all baseline models, demonstrating its superior ability in recognizing sequence entities. However, when comparing Pair F1 scores, which assess the accuracy of predicting relationships between entities, we observe more variation across models. While UniVIE and other baselines perform well, our model remains competitive, particularly in the "LayoutLMv3 Chinese

Base" version, which excels in both entity recognition and relation extraction. Analyzing the four versions of our model reveals valuable insights. The "Layout embedding only" version shows that spatial information alone provides a solid foundation for both tasks, though it benefits from additional textual context. The "Word2vec embedding" version shows slight improvements, indicating that word-level embeddings contribute but are not sufficient on their own. The "Byte pair encoding" version, with a significant boost in Pair F1, demonstrates its ability to handle complex or rare entity relations more effectively. Finally, the "LayoutLMv3 Chinese Base" version, which integrates both layout and language features, achieves the best overall results, highlighting the importance of combining both modalities for robust performance in entity recognition and relation extraction tasks. Figure 4 illustrates some qualitative results of node label diffusion.

## 4.4 REALISTIC VIE PERFORMANCE TEST ON CORD

**Evaluation metrics.** Node coordinate generation is an unconditional generation task, where the positions and sizes of multiple boxes are generated directly. Edge coordinate generation is a conditional task, where the conditions are the node coordinates and their feature representations from the node coordinate diffusion module. Since the dataset contains ground truth annotations, we evaluate the accuracy of the generated edge coordinates using F1 scores.

**Quantitative and qualitative results.** Table 2 presents the performance of various models on the CORD dataset for the realistic VIE task, comparing results based on SER F1, RE F1, and Pair F1 metrics. In terms of SER F1, our model reaches an impressive score of 97.11%, surpassing all baseline models and achieving state-of-the-art performance in sequence entity recognition. This demonstrates the strength of our approach in identifying entities accurately from visual documents. However, when we turn to Pair F1, which is a more stringent metric that evaluates the model's ability to predict relations between entities using the predicted labels from SER, our model shows a score of 96.67%. Although this is slightly lower than the highest score (98.32% from DocFormer), it is still highly competitive, especially considering the increased complexity of the task. The fact that our model performs so close to state-of-the-art under this more challenging metric highlights its robustness in handling complex relationships between entities.

As shown in Figure 5, UniVIEDM successfully models the spatial distribution patterns of node coordinates across different types of layout samples, and appropriately selects the number of nodes for each layout type. For edge coordinates, UniVIEDM also provides accurate estimates regarding the presence and quantity of edges, and is capable of generating precise edge coordinates solely based on node coordinates. This indicates that our node and edge coordinate diffusion models effectively learn the probability distributions of node and edge coordinates in the plane. Notably, our model successfully leverages the jump-diffusion process to address the challenges posed by trans-dimensional characteristics and non-sequential properties of the data. Moreover, we observe that UniVIEDM, relying solely on layout modality, is able to address critical aspects of the relation extraction (RE) task, thus fully leveraging the potential of layout modalitsy in VIE tasks.

## 4.5 CONCLUSION

In this paper, we introduced UniVIEDM, a unified diffusion model for Visual Information Extraction (VIE) tasks, addressing key challenges in extracting named entities and relationships from visually rich documents. By leveraging plane graph structures and integrating layout and textual features, our model unifies the traditionally separate subtasks of line grouping, semantic entity recognition (SER), and relation extraction (RE) into a single framework. The introduction of jump-diffusion for coordinate generation and continuous-time Markov chain (CTMC) processes for label prediction allows UniVIEDM to effectively handle the complexities of trans-dimensional and mixed-type data.

Extensive experiments on public datasets, including SIBR and CORD, demonstrate that UniVIEDM achieves state-of-the-art performance in SER F1 while maintaining competitive results in the more stringent Pair F1 metric. Our model's ability to integrate layout and text features proves crucial in achieving robust and accurate VIE results, as shown by its superior performance across all tasks. The results validate the effectiveness of our unified approach, suggesting that it can overcome the limitations of previous systems that treat these tasks separately.

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
