# OpenReview forum: "UniVIEDM: A Diffusion Model to Unify Visual Information Extraction Subtasks"
_ICLR.cc/2025/Conference — Submitted to ICLR 2025_

### Official Review · Reviewer_kAUF · 2024-10-31

**Soundness:** 2
**Presentation:** 3
**Contribution:** 2
**Rating:** 5
**Confidence:** 4

**Summary:**

The paper proposed to use graph based diffusion models to model the generative process of document and forms and subsequently generatively model the label formulation for graph nodes and edges, unifying 3 mainstream document understanding tasks.

The proposed paper aims to address the limitation of related works that lacks unification of VIE tasks and also adopts state-of-the-art graph based diffusion method for document understanding task. The paper is very well written and overall a great reading experience.

**Strengths:**

[1] Generative modeling on graphs using diffusion formulation should be praised and acknowledged
- This is the core novelty of this paper. Indeed non trivial to work on variable shaped graph data (N nodes and N edges are changing).
- It is a good starting point to research on document and form generation using graph DMs.

[2] Solid writing and great related work walkthrough.
- Related works are nicely introduced.
- Eqn 1->6 is neat and clearly correspond to different model design in this paper.
- Excellent work overall.

[3] Unification
- This paper proposed a framework that unifies three main stream tasks, which is a plus considering many related work only does one or two (albeit some related work does three as well).

**Weaknesses:**

[1] Novelty
- Novelty is limited in this paper.
- The core methods for graph diffusion is not proposed by this paper but rather in Campbell et.al. 2024, and is used in all DM modules in various stages in this paper.
- There is no solid proof that using diffusion model vs discriminative GCNs proves significant benefits. Please see below points on "efficiency" and "model design".
- Overall my impression is that the paper uses DM without solid motivation and proven benefits in either efficiency or super strong metric wins, it applies DM for the sake of applying it. **AND** the graph DM method and formulation is not original in this paper.

[2] Plain graph formulation.
- This is not entirely new or novel. See the beta-skeleton graph on ROPE paper "ROPE: Reading Order Equivariant Positional Encoding for Graph-based Document Information Extraction".
- Both graph formulation requires first a OCR engine to generate raw nodes, and yet beta graph possesses better properties to avoid unnecessary edge formulations.

[3] Claims on efficiency
- One big motivation found in this paper to apply diffusion modules on graph is to be "more efficient than quadratic compute requirement found in modeling relationships in graphs". I am not convinced by this claim at all. Consider below A, B methods:
- A: GCN based methods work on N nodes and N^2 edges and indeed the complexity is bounded by O(N^2). This can be ROPE or many cited related work in this paper.
- B: the graph diffusion frame work does not escape from this complexity, you need to predict epsilons for the full graph structure in one functional diffusion model inference, and you also need to iteratively run the DM to fully recover the final label predictions.
- When the underlying GCN or GNN is the same in both A and B, A requires 1 pass of GCN inference. B requires multiple passes due to denoising diffusion modules.
- Therefore, I am super suspicious on this efficiency claim. Meanwhile, there is no analysis nor experiments to support the DM formulation proposed in this paper provides any efficiency gain over related GCN or discriminative modeling work.

[4] Model design
- 1st stage: it seems the core need for this model is not to generate novel documents or layouts, but in this work it is to generate node level features in the latent space? How exactly is the feature from DM model is computed? Why do we not use traditional GCN applied in so many related works to do feature abstraction? Any comparisons?
- 2nd stage edge diffusion, same question as above.
- Last stage node and edge label prediction diffusion: this I can somehow side with, and it seems using diffusion model might give a slight boost over simpler and cheaper GCN methods. But I still suspect the DM model used here is not in theory cheaper than a corresponding GCN since the DM module here operates on graph and relies on similar underlying model arch. On top of that you need multiple denoising steps. The small gains cannot easily justify the computational cost incurred by the many diffusion steps.
- **The generative power of DM is not realized by this design**. This work is using generative models to perform discriminative tasks: the generation cannot generate novel forms or documents, but rather generate labels on existing layout (3rd stage model). The first two stage models are some what interesting but its capacity is limited to generate layouts only and there are lots of existing works already achieving this (related work section in this paper).

**Questions:**

Weakness [3] Efficiency
- How many diffusion steps?
- Experiment request: please find the most competitive GCN based approach in your comparison table, and use the same underlying GCN arch for diffusion modules here. Then conduct run time comparisons.

Weakness [4] model design
- Experiment request: do not use DM for stage 1 and 2 but use a pretrained GCN on similar tasks to extract node and edge features. Then train last stage DM to predict labels. I am super inclined that the designed 1,2 stage won’t add much benefit here.
- Please consider discuss paper ROPE mentioned above and highlight your method's pros/cons over it.

If the authors can provide proof that this method gives notable efficiency gains and that each stage of diffusion model is worthy of its own, I would be very impressed and willing to change my rating.

---

### Official Review · Reviewer_DJ6K · 2024-11-03

**Soundness:** 2
**Presentation:** 2
**Contribution:** 2
**Rating:** 1
**Confidence:** 5

**Summary:**

This paper introduces UniVIEDM, a new method for Visual Information Extraction (VIE) that seeks to integrate the three primary subtasks, line grouping, Semantic Entity Recognition (SER), and Relation Extraction (RE), into a unified diffusion model framework. Rather than addressing these subtasks sequentially or through distinct models, UniVIEDM utilizes the concept of plane graphs to depict document layouts and formulates all three subtasks as problems of plane graph generation. This strategy presents an effective solution to the computational complexities and error propagation challenges.

**Strengths:**

1. UniVIEDM integrates the three subtasks of VIE, line grouping, SER, and RE, into a single model, which may minimize computational complexity and streamline the overall architecture.

2. UniVIEDM demonstrates competitive performance across multiple public datasets, showcasing its efficacy in real-world applications.

3. The use of jump-diffusion and CTMC allows UniVIEDM to address the issues of varying data dimensions and insensitivity to the generation order.

**Weaknesses:**

1. The writing of the paper requires improvement. There is a lack of quantitative analysis supporting the main claim regarding the reduction in computational complexity. As it stands, the current version is not convincing to me.

2. The novelty of the paper appears to be limited, primarily presenting a straightforward combination of existing methods, such as diffusion models and continuous-time Markov chains (CTMC). Additionally, I find that the current version lacks a compelling rationale for employing the diffusion model to address the VIE tasks.

3. The paper lacks an ablation study to evaluate the contribution of each component of UniVIEDM. Such an analysis would offer valuable insights into the significance of various elements of the model.

4. The paper frequently references details, such as the network architecture design, that are said to be included in the appendix. However, I was unable to locate the appendix, which is quite unprofessional.

**Questions:**

See "weaknesses"

---

### Official Review · Reviewer_qVSu · 2024-11-04

**Soundness:** 2
**Presentation:** 1
**Contribution:** 2
**Rating:** 3
**Confidence:** 5

**Summary:**

This paper addresses the task of Visual Information Extraction (VIE) for documents by introducing an approach called Unified VIE Diffusion Model (UniVIEDM). The proposed approach tries to solve the entity detection and relationship prediction problem simultaneously. It does this by considering the layout of a document as a plane graph consisting of text entities as nodes and relationships/connections between them as the edges. The paper describes the pre-training and fine-tuning methodologies for training the model and shows results on multiple benchmarks.

**Strengths:**

The strength of the paper is the idea of a unified approach for entity detection and linking for VIE tasks in document processing. The paper motivates the approach well and the authors select multiple datasets to benchmark the proposed approach.

**Weaknesses:**

Missing details, explanations, references are the biggest weakness of the paper. The paper will be significantly improved if the authors can address the following:

1. Lines 272-273 say "Their network design can be found in the appendix". This is important information and needs to be included in the main body of the paper.

2. In lines 205-206, what does "2+ types of edge labels" mean?

3. The Experiments (Section 4) section needs to include more detail and ablation studies to explain the utility and functioning of the proposed approach. Currently there is no analysis or ablation in the experiments section. This makes it impossible to really understand how and why does the proposed approach function as it does.

4. There have been some prior methods which try to combine entity detection and linking into a unified approach. An example of this is [1] which seems to achieve a higher performance on the CORD dataset (Table 3) than the proposed approach. The authors should discuss such prior approaches and compare against them.



[1] Liao, Haofu, Aruni RoyChowdhury, Weijian Li, Ankan Bansal, Yuting Zhang, Zhuowen Tu, Ravi Kumar Satzoda, R. Manmatha, and Vijay Mahadevan. "Doctr: Document transformer for structured information extraction in documents." In Proceedings of the IEEE/CVF International Conference on Computer Vision, pp. 19584-19594. 2023.

**Questions:**

Please see weaknesses section above.

---

### Official Review · Reviewer_pALB · 2024-11-05

**Soundness:** 3
**Presentation:** 4
**Contribution:** 3
**Rating:** 3
**Confidence:** 3

**Summary:**

This paper presents a novel integrated framework for visual information extraction (VIE). Unlike most VIE algorithms, which process three sequential steps independently, the proposed algorithm integrates these modules into a unified, diffusion-based model. By effectively combining entity relationships, jump-diffusion processes, and a continuous-time Markov chain, the model significantly reduces the computational complexity often encountered in joint VIE models. The algorithm was validated using official VIE benchmark datasets, demonstrating state-of-the-art performance in SER F1 and Pair F1 metrics.

**Strengths:**

The introduction and related work sections are well-organized, providing readers with a clear understanding of the topic. Furthermore, the contributions and target issues are clearly articulated, highlighting the novelty of the proposed algorithm. The experimental settings are thoroughly described, ensuring that the validation experiments can be easily reproduced. The framework incorporates jump diffusion models, which are specifically tailored to enhance its performance and effectiveness.

**Weaknesses:**

1. Evaluation for Computational Complexity
   The authors claim a significant reduction in computational complexity in this paper; however, in the experiments, they merely perform a performance comparison. Thus, since the contribution to computational complexity has not been empirically demonstrated, it is difficult to emphasize this claim. It is necessary to experimentally show the effect of reduced computational complexity in the proposed algorithm, such as by comparing training times or the number of relationship combinations considered within a fixed train iteration.

2. Impact of the Scaling Issue
   The authors mention the scaling issue as the main challenge in existing joint model algorithms, yet they only discuss how this issue affects computational complexity without detailing its practical impact. If computational complexity is high but does not significantly affect training, the contribution of the proposed algorithm could be diminished, as the effect of addressing this issue may be minimal. It is crucial to confirm and examine how reducing computational complexity can resolve an important issue in existing joint models.

3. Insufficient Comparison with Joint Models
   In Table 1, only UniVIE represents joint models, and Table 2 does not include any joint models. This limits a fair comparison to demonstrate the effectiveness of the proposed algorithm against prior joint models.

4. Missing Metrics in RE F1
   In Tables 1 and 2, the RE F1 metric for the proposed algorithm is missing. In particular, only Pair F1 is measured for the proposed algorithm in Table 2, leaving the corresponding cells for previous algorithms blank, making it difficult to clearly compare algorithm performance. As SER F1 alone does not fully represent the performance of VIE, the blank cells should be filled to enable a fair and transparent comparison across algorithms.

**Questions:**

How can the authors empirically demonstrate the claimed reduction in computational complexity?
What impact does the scaling issue have on joint models, and how does the proposed algorithm effectively address it?
How can missing metrics and the lack of comprehensive comparison with joint models affect the fairness of the evaluation?

---

### Meta-Review · Area_Chair_HhrB · 2024-12-17

**Metareview:**

This paper presents a Unified VIE Diffusion Model (UniVIEDM) for all the tasks of visual information extraction (VIE). The paper initially got four negative scores.


The main strengths include: 1) well-organized; 2) experimental settings are thoroughly described; 3) the proposed method shows competitive performance across several datasets;

However, there are several drawbacks raised by the reviewers: 1)  missing computational evaluation; 2) insufficient comparison with joint models; and 3) missing details, explanations and references.

The authors did not submit a rebuttal. The reviewers have viewed the comments of the others and decided to keep their original ratings. Considering the drawbacks raised by the reviewers, the AC thinks this paper cannot meet the requirement of ICLR at this point and thus regrets to recommend rejection.

**Additional Comments On Reviewer Discussion:**

The authors did not submit a rebuttal. The reviewers have viewed the comments of the others' and confirmed the drawbacks raised in the first round.

---

### Decision · Program_Chairs · 2025-01-22

Reject